# Towards Noise-Tolerant Speech-Referring Video Object Segmentation: Bridging Speech and Text

**Xiang Li**[1]   **Jinglu Wang**[2]   **Xiaohao Xu**[3]   **Muqiao Yang**[1]
**Fan Yang**[4]   **Yizhou Zhao**[1]   **Rita Singh**[1]   **Bhiksha Raj**[1,5]

[1] Carnegie Mellon University   [2] Microsoft   [3] University of Michigan Ann Arbor
[4] Ohio State University   [5] Mohamed bin Zayed University of Artificial Intelligence
xl6@andrew.cmu.edu

## Abstract

Linguistic communication is prevalent in Human-Computer Interaction (HCI). Speech (spoken language) serves as a convenient yet potentially ambiguous form due to noise and accents, exposing a gap compared to text. In this study, we investigate the prominent HCI task, Referring Video Object Segmentation (R-VOS), which aims to segment and track objects using linguistic references. While text input is well-investigated, speech input is under-explored. Our objective is to bridge the gap between speech and text, enabling the adaptation of existing text-input R-VOS models to accommodate noisy speech input effectively. Specifically, we propose a method to align the semantic spaces between speech and text by incorporating two key modules: 1) Noise-Aware Semantic Adjustment (NSA) for clear semantics extraction from noisy speech; and 2) Semantic Jitter Suppression (SJS) enabling R-VOS models to tolerate noisy queries. Comprehensive experiments conducted on the challenging AVOS benchmarks reveal that our proposed method outperforms state-of-the-art approaches.

## 1 Introduction

Recent advances in vision-language learning have significantly advanced Human-Computer Interactions (HCI). A demanding task within HCI is referring video object segmentation (R-VOS), which involves segmenting and tracking objects in videos based on textual references. The successful development of R-VOS techniques has paved the way for diverse real-world applications such as video editing (Li et al., 2022d) and augmented reality (Huang et al., 2022). Notably, recent R-VOS methods (Luo et al., 2023) have shown unprecedented progress, propelled by the rapid advancement of multimodal foundation models such as CLIP (Radford et al., 2021). These R-VOS models enable various text-referred scenarios, allowing referring segmentation for generalized textual expressions even in complex visual scenes.

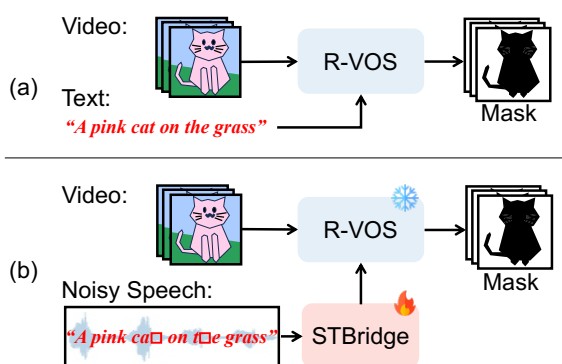

Figure 1: (a) Referring video object segmentation (R-VOS) employs text as a query for segmentation. (b) Speech-referring video object segmentation. Compared to *written language* (text), *spoken language* (speech) is a more noisy form, potentially involving greater information loss and disturbance due to background noises. We propose a plug-and-play *STBridge* module, which seamlessly extends a frozen text-conditioned R-VOS model to accommodate noisy speech inputs.

However, a more challenging scenario arises in the prevalent speech dialogue system, where the aim is to refer to specific targets using spoken language, i.e., speech. The inherent nature of speech introduces vulnerabilities to disturbances from background noises (sound except for the referring speech). Consequently, crucial information within the spoken content can be distorted or even lost, which poses extra challenges in maintaining effective segmentation when referring to targets verbally. Though previous R-VOS methods achieve remarkable performance with textual queries, their performance with real-world spoken language is rarely discussed. Hence, it is crucial to develop an effective approach to bridge the text and speech to adapt well-trained R-VOS methods (frozen) for speech inputs.

Yet, bridging speech to R-VOS methods introduces new challenges. A straightforward solution involves utilzing automatic speech recognition (ASR) (Li et al., 2022a) to convert speech

to text, followed by text-conditioned referring segmentation. However, this can result in suboptimal performance for two primary reasons. (1) *Noise in language queries of R-VOS*. Existing R-VOS models rely on clean text-video pairs, wherein the textual expression unambiguously identifies the target object. Nonetheless, information extracted from speech may be incomplete or distorted due to background noise and ASR errors, leading to inadequate references to the target object. To maintain robust segmentation quality, it is crucial to adapt R-VOS models to handle perturbed referring queries. (2) *Noise in speech understanding*. In practice, speech and background noise are closely intertwined, making it difficult to accurately comprehend semantic information from speech. Considering the diverse types of noise, an effective noise-tolerant speech understanding approach is vital for achieving robust speech-referring video object segmentation.

In this paper, we present STBridge, a novel approach that enables R-VOS models trained on **clean** text-video pairs to adapt to **noisy** speech as referring guidance, maintaining robust performance even amidst background noises. As illustrated in Fig. 1, the proposed STBridge links the well-trained R-VOS model with speech input, incorporating two core considerations to improve the model's robustness: (1) enhancing the well-trained R-VOS model to accept incomplete guidance, and (2) providing the noise-tolerant capability for speech understanding. On the one hand, we introduce a semantic-jitter suppression (SJS) module to help the R-VOS model understand noisy information from referring guidance. The SJS module generates object queries with randomly jittered textual features, allowing the model to learn from incomplete referring guidance under proper supervision. On the other hand, we introduce a noise-aware semantic adjustment (NSA) module, which generates noise-adaptive filters to enhance the speech representation. This differs from traditional speech enhancement, as it focuses solely on encoded semantics during speech understanding, while discarding low-level information, i.e., waveforms.

We further introduce a slack semantic alignment to align text and speech queries, enabling the integration of speech input with well-trained R-VOS models. Notably, our method incorporates additional modules without any retraining of R-VOS models, which is essential for numerous real-world applications. In summary, our contributions are as follows:

- We propose STBridge, a novel approach to bridge speech input to referring segmentation models, enabling segmenting objects with spoken language.

- We introduce semantic-jitter suppression and noise-aware semantic adjustment modules to enable the noise-tolerant capability for speech queries.

- We conduct extensive experiments on speech-referring segmentation benchmarks and the results of which show our approach performs favorably over prior arts.

## 2 Related Works

**Video segmentation.**  Video segmentation (Wang et al., 2021c; Li et al., 2022b,c, 2023a; Yan et al., 2023; Li et al., 2023c) is a fundamental task to enable video editing. Semi-supervised video object segmentation (VOS) which leverages a first-frame mask to assign the target object is among the most popular video segmentation tasks due to its high segmentation quality.  Some recent works (Yang et al., 2020, 2021) propagate masks by exploring matches among adjacent frames. Space-Time-Memory networks (STM) (Oh et al., 2019) builds a memory bank for matches. Several works follow the paradigm used in STM and improve the memory construction policy (Xie et al., 2021; Liang et al., 2020; Wang et al., 2021a) or enhance the memory reading strategy (Cheng et al., 2021a; Seong et al., 2020; Hu et al., 2021; Cheng et al., 2021b; Seong et al., 2021; Yang et al., 2021). Recently, Yan et al.(Yan et al., 2023) introduced a two-shot setting for VOS tasks which enables high-performance segmentation with limited annotated frames. Since the VOS task is primarily used for video editing which requires human involvement, to reduce the labor in assigning the target object, referring video object segmentation (R-VOS) is introduced.  Specifically, R-VOS aims to segment an object in a video sequence given a linguistic description as the query. ReferFormer (Wu et al., 2022) and MTTR (Botach et al., 2022) are two pioneering works that utilize transformers to decode or fuse multimodal features. Recently, $R^2$-VOS (Li et al., 2023b) introduces a cyclic structural consistency to enhance the robustness of R-VOS. And

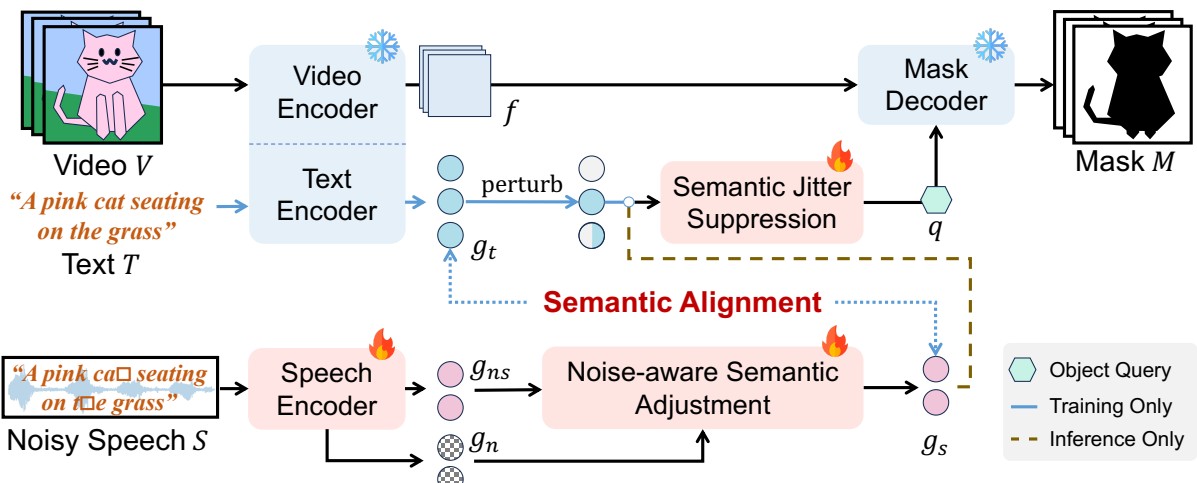

Figure 2: **Overview of STBridge**. We extend frozen R-VOS model (in blue) with trainable STBridge (in red) modules to adapt to noisy speech as input. (I) **Training**: A speech encoder is utilized to extract noisy speech $g_{ns}$ and noise $g_n$ embeddings from noisy speech $S$. On one hand, a *noise-aware semantic adjustment (NSA)* module is utilized to mitigate the noise influence, which derives the cleaner speech embedding $g_s$. On the other hand, to enhance the noise-tolerant capability of R-VOS model, we first generate perturbation to the text embedding $g_t$ and then equip a semantic jitter suppression (SJS) module to suppress the noises. Moreover, semantic alignment constraints are introduced to align the text $g_t$ and speech $g_s$ embeddings. (II) **Inference**: After aligning the text and speech embeddings during training, we can directly discard the text branch and leverage speech embedding $g_s$ as the input to the SJS module.

OnlineRefer (Wu et al., 2023) employs the query propagation module to enable the online R-VOS.

**Spoken language understanding.** Spoken language, *i.e.*, speech, enables a more natural way for humans to refer to a certain object than using text-based language. Thanks to the emergence of datasets with paired images and speech, *e.g.*, Flicker8K (Harwath and Glass, 2015) and AVOS (Pan et al., 2022), more works (Chrupała, 2022; Harwath et al., 2020; Kano et al., 2021; Seo et al., 2023) started to research on the representation of speech and explore the synergy between speech and other modalities, *e.g.*, image and video. For example, LAVISH (Lin et al., 2023) incorporates a small set of latent tokens to align the visual and audio representation, and VisualVoice (Gao and Grauman, 2021) conducts speech separation with the speaker's facial appearance as a conditional prior. Later, research on speech has also moved towards finer granularity tasks. Some works (Lei et al., 2021) focus on the mono-modal impact of speech to study the subtle semantic information of spoken language to better understand human speech, while others (Jiang et al., 2021) study how to introduce the knowledge of speech understanding to create more natural human-computer interaction applications, *e.g.* talking head (Hwang

et al., 2023; Li et al., 2023c; Qu et al., 2023).

## 3 Method

To ground objects verbally, we start from a frozen text-referring video object segmentation (R-VOS) model (shown as blue modules in Fig. 2), including frozen video-text encoders and a mask decoder. We introduce a speech encoder, a semantic jitter suppression (SJS) module, a noise-aware semantic adjustment (NSA) module, and a semantic alignment constraint to bridge text and speech (shown as pink modules in Fig. 2). During training, STBridge leverages video $V$, text $T$, and noisy speech $S$ triplets to align the query spaces between text and speech. Thereafter, we can discard the text branch and directly query the objects with speech during inference.

### 3.1 Encoders

**Frozen video and text encoders.** We consider a generic referring segmentation framework that equips a video encoder $\mathcal{E}_v$ and a text encoder $\mathcal{E}_t$ to extract visual and textual features. Let us denote the extracted visual feature as $f = \mathcal{E}_v(V) \in \mathbb{R}^{C_v \times L_v \times H \times W}$ and extracted text embeddings as $g_t = \mathcal{E}_t(T) \in \mathbb{R}^{C \times L_t}$, where $C_v, C$ and $L_v, L_t$ are the channel and length of visual and text embeddings respectively. We freeze the video and text

encoders during both training and inference.

**Speech encoder.** We leverage a transformer-based speech encoder, Wav2Vec2 (Baevski et al., 2020) to extract speech features. We additionally augment two linear layers on top of the last hidden state of Wav2Vec2 to predict noise type. Thereby, each speech embedding corresponds to a noise embedding to describe the noise information. We denote the extracted noisy speech embedding as $g_{ns} \in \mathbb{R}^{C \times L_s}$ and noise embedding as $g_n \in \mathbb{R}^{C \times L_s}$. $C$ and $L_s$ are the channel and length of embeddings.

## 3.2 Semantic Jitter Suppression

To equip the R-VOS model, which is typically trained on clean data samples, with the noise-tolerance capability, we first mimic noisy text embeddings $g'_t$ by applying semantic jitters to the original text embeddings $g_t$. After that, we introduce a learnable semantic jitter suppression block $\varphi(\cdot)$ to suppress the jitter and generate proper object query $q$ for the following mask decoding.

Specifically, we implement the semantic jitter with a linear perturbation function where $g'_t = m \circ g_t + \delta$. Here, $m \in \{0, 1\}^{C \times L_t}$ is a binary masking operation at either word-level (along $L_t$ dimension) or channel-level (along $C$ dimension); $\delta \in \mathbb{R}^{C \times L_t}$ is a random noise; $\circ$ denotes the Hadamard product. Besides, the jitter suppression block is constructed by cascading a transformer encoder and a global average pooling layer which pools along the word dimension. Formally, the final object query $q \in \mathbb{R}^{C \times 1}$ can be generated as

$$q = \varphi(m \circ g_t + \delta). \qquad (1)$$

## 3.3 Noise-aware Semantic Adjustment

We introduce noise-aware semantic adjustment (NSA) to adjust inaccurate semantics introduced by noises, which consists of two components: a bi-directional cross-attention for noise-speech interaction and a noise-guided modulation for speech embedding adjustment.

**Bi-directional cross-attention (BCA).** In BCA, Noise-to-Speech (N-S) and Speech-to-Noise (S-N) cross-attention layers are involved to compute noise-aware speech embeddings $g'_n$ and speech-aware noise embedding $g'_n$. Formally, they take the form:

$$h_{n \to s} = \text{Softmax}\left(Q_n^{\text{T}} K_s / \sqrt{d}\right) V_s \qquad (2)$$

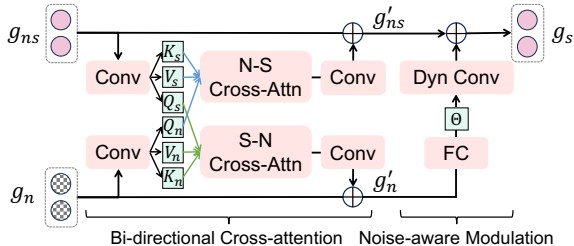

Figure 3: Illustration of Noise-aware Semantic Adjustment. Noisy speech and noise embeddings, *i.e.*, $g_{ns}$ and $g_n$, first interact with each other via the bi-directional cross-attention mechanism. Then, the fused noise embedding $g'_n$ is used to modulate the fused speech embedding $g'_{ns}$ to make it more noise-aware.

$$h_{s \to n} = \text{Softmax}\left(Q_s^{\text{T}} K_n / \sqrt{d}\right) V_n, \qquad (3)$$

where $h_{n \to s}$ and $h_{s \to n}$ are outputs of N-S and S-N attention. $K$, $Q$, and $V$ are derived by applying linear projections on the original speech or noise embedding. $d$ is the dimension of $K$ and $Q$. The $h_{n \to s}$ and $h_{s \to n}$ are fused back to their paths with residual connections (He et al., 2016). We denote the fused embeddings as $g'_{ns}$ and $g'_n$.

**Noise-guided modulation (NGM).** To incorporate noise information into speech embeddings, we propose a noise-guided feature modulation with channel-wise attention (Tian et al., 2020). Different from attention in BCA acting along the time dimension, channel-wise attention directly acting on feature channels is more efficient to exploit semantically meaningful correlations (Wang et al., 2021b; Tian et al., 2020), especially for instance-level correlations (Tian et al., 2020; Cao et al., 2020; Bolya et al., 2019). Given the speech-aware noise embedding $g'_{ns}$, we first apply a fully connected layer on it to form the dynamic filters $\Theta = \{\theta_i\}_{i=1}^{L_s}$. Here, each filter $\theta_i \in \mathbb{R}^{C \times 1}$ represents the noise information for each timestep and modulates the speech embeddings according to their category and amplitude. Then we utilize channel-wise attention to modulate the noise-aware speech feature $g'_s$, which is given by:

$$g_{s|n} = \Theta \circ g'_{ns} \qquad (4)$$

where $g_{s|n}$ is the modulated speech embeddings and $\circ$ represents Hadmard product. We fuse the $g_{s|n}$ back to $g'_{ns}$ with a residual connection, which derives the final output $g_s = g'_{ns} + g_{s|n}$.

## 3.4 Frozen Mask Decoder

Referring segmentation methods typically leverage a query-based mask decoder $\mathcal{D}(q, f)$ (Wu et al.,

2022; Botach et al., 2022) that takes an object query $q \in \mathbb{R}^{C \times 1}$ encoding the object information and a video feature $f$ as inputs to predict the object masks $M \in \mathbb{R}^{N \times L_v \times H_o \times W_o}$, object bounding boxes $B \in \mathbb{R}^{N \times L_v \times 4}$ and confidence scores $S \in \mathbb{R}^{N \times L_v \times 1}$ across video frames. $N$ is the object candidate number. Here, we omit the detailed structure (available in Appendix) for simplicity. It is worth mentioning that the object query for the decoder is simply an averaged text embedding for recent popular R-VOS methods, which takes the form: $q = \text{pool}(g_t)$. The well-trained mask decoder $\mathcal{D}$ keeps frozen in our method.

## 3.5 Training Objectives

We utilize a semantic alignment loss $\lambda_{align}$ to align speech and text queries, a noise classification loss $\mathcal{L}_{noise}$ to facilitate speech understanding, and a segmentation loss $\mathcal{L}_{match}$ to segment objects:

$$\mathcal{L} = \lambda_{align}\mathcal{L}_{align} + \lambda_{noise}\mathcal{L}_{noise} + \mathcal{L}_{match} \quad (5)$$

where $\lambda_{align}$ and $\lambda_{noise}$ are constants.

**Semantic alignment.** To bridge the text and speech queries, we conduct semantic alignment between text and speech embeddings. As the object query $q$ requires sentence-level semantics (each sentence describes one object), it is not necessary to enforce a tight sequence-to-sequence alignment between text and speech embeddings (Tan et al., 2023). Instead, we align text and speech embeddings with a loose constraint. Specifically, given a text embedding $g_t$ and a speech embedding $g_s$, we first pool them among word and time dimensions correspondingly. After that, an alignment constraint is applied between them

$$\mathcal{L}_{align} = \|\text{pool}(g_t) - \text{pool}(g_s)\|_2 \quad (6)$$

where $\|\cdot\|_2$ is the L$_2$-Norm.

**Noise classification.** We augment the clean speech with different categories of audio, e.g., dog barking, as noise. We apply a noise classification head on top of noise embedding $g_n$ to predict noise categories. Let us denote the predicted probabilities as $p \in \mathbb{R}^{N_c \times 1}$ and the ground truth class as $c$, where $N_c$ is the noise type number. The noise classification loss $\mathcal{L}_{noise}$ can be computed as

$$\mathcal{L}_{noise} = -\log p[c] \quad (7)$$

where $p[c]$ denotes the probability of class $c$.

**Object segmentation.** Following the object segmentation methods (Wu et al., 2022; Wang et al., 2021c), we assign each mask prediction with a ground-truth label and then apply a set of loss functions between them to optimize the segmentation mask quality. Given a set of predictions $y = \{y_i\}_{i=1}^{N}$ and ground-truth $\hat{y} = \{\hat{B}_l, \hat{S}_l, \hat{M}_l\}_{l=1}^{L_v}$ where $y_i = \{B_{i,l}, S_{i,l}, M_{i,l}\}_{l=1}^{L_v}$, we search for an assignment $\sigma \in \mathcal{P}_N$ with the highest similarity where $\mathcal{P}_N$ is a set of permutations of $N$ elements. The similarity can be computed as

$$\mathcal{L}_{match}(y_i, \hat{y}) = \lambda_{box}\mathcal{L}_{box} + \lambda_{conf}\mathcal{L}_{conf} \\ + \lambda_{mask}\mathcal{L}_{mask} \quad (8)$$

where $\lambda_{box}$, $\lambda_{conf}$ and $\lambda_{mask}$ are constant numbers to balance the losses. Following previous works (Ding et al., 2021; Wang et al., 2021c), we leverage a combination of Dice (Li et al., 2019) and BCE loss as $\mathcal{L}_{mask}$, focal loss (Lin et al., 2017b) as $\mathcal{L}_{conf}$, and GIoU (Rezatofighi et al., 2019) and L1 loss as $\mathcal{L}_{box}$. The best assignment $\hat{\sigma}$ is solved by the Hungarian algorithm (Kuhn, 1955).

## 3.6 Inference

During inference, we only keep speech and video as inputs. We first pool the speech embedding $g'_s$ into a fixed size and then utilize it to replace the text embedding $g_t$. Thereby, noisy speech can replace text to query the visual object. Please note that the text branch stays functional as we froze the R-VOS model during the training of STBridge.

# 4 Experiment

## 4.1 Datasets and Metrics

**Datasets.** We conduct experiments on the large-scale speech-referring video object segmentation dataset, AudioGuided-VOS (AVOS) (Pan et al., 2022) which augments three R-VOS benchmarks with speech guidance: Ref-YoutubeVOS (Seo et al., 2020), A2D-sentences (Xu et al., 2015) and JHMDB-sentences (Jhuang et al., 2013). Specifically, it involves 18,811 pairs of video sequences and speech audio, which is divided into the training, validation, and test set in a ratio of 0.75, 0.1, and 0.15, respectively. The AVOS test set only contains Ref-YoutubeVOS samples. The A2D-sentences and JHMDB-sentences test sets are evaluated on their original test splits with speech as queries. Based on the AVOS dataset, we synthesize noisy speech by combining randomly picked audio from

| Method | Query | No Noise | | Noise (30 dB) | | Noise (20 dB) | | Noise (10 dB) | |
|---|---|---|---|---|---|---|---|---|---|
| | | $\mathcal{J}$ | $\mathcal{F}$ | $\mathcal{J}$ | $\mathcal{F}$ | $\mathcal{J}$ | $\mathcal{F}$ | $\mathcal{J}$ | $\mathcal{F}$ |
| Text-referring Video Object Segmentation (for reference) | | | | | | | | | |
| ReferFormer (Wu et al., 2022) | Text | 66.3 | 68.1 | - | - | - | - | - | - |
| MTTR (Botach et al., 2022) | Text | 63.2 | 64.7 | - | - | - | - | - | - |
| Noisy Speech-referring Video Object Segmentation | | | | | | | | | |
| ReferFormer (Wu et al., 2022) | Text (ASR) | 56.1 | 60.7 | 55.3 | 58.9 | 53.3 | 58.1 | 50.3 | 56.4 |
| MTTR (Botach et al., 2022) | Text (ASR) | 52.4 | 56.8 | 51.8 | 56.5 | 49.4 | 55.4 | 47.8 | 53.4 |
| **STBridge (Ours)** | Speech | **63.7** | **67.4** | **63.5** | **67.1** | **62.1** | **66.1** | **59.9** | **64.8** |

Table 1: **Quantitative results for noisy speech-referring object segmentation in videos.** The noise loudness (dB) is measured by the signal-noise ratio (SNR) to the clean speech.

Audioset (Gemmeke et al., 2017). Specifically, a noise ranging from 0 to 40 dB signal-noise ratio (SNR) to the clean speech is sampled during training. For validation and testing, we create noisy speech under 10 dB, 20 dB, and 30 dB SNR for comprehensive evaluation.

**Metrics.** We leverage the region similarity $\mathcal{J}$ and contour accuracy $\mathcal{F}$ (Pont-Tuset et al., 2017) metrics for the evaluation of speech-referring video object segmentation. The overall evaluation metric $\mathcal{J}\&\mathcal{F}$ is the average of $\mathcal{J}$ score and $\mathcal{F}$ score. Both the $\mathcal{J}\uparrow$ and $\mathcal{F}\uparrow$ scores are the larger the better.

### 4.2 Implementation Details

We implement our method in PyTorch. Without losing generality, we leverage ReferFormer (Wu et al., 2022) as our frozen R-VOS model (can be replaced with any query-based model). We train our model for 2 epochs with a learning rate of 1e-4. All experiments are run on 8 NVIDIA V100 GPUs. We adopt $\mathrm{batchsize}$ 8 and an AdamW (Loshchilov and Hutter, 2017) optimizer with weight decay $5 \times 10^{-4}$. Images are cropped to have the longest side 640 and the shortest side 360 during training and evaluation. In Eq. 1, we utilize the random noise $\delta \sim \mathrm{Uniform}(-0.5, 0.5)$ and a masking ratio of 0.1 for $m$ as default. Please refer to the Appendix for more details.

### 4.3 Quantitative Results

**Segmentation with clean speech.** Table 2 compares the proposed STBridge with previous methods using the ResNet-50 (He et al., 2016) backbone. To better analyze the performance of STbridge, we introduce two popular R-VOS baselines (with text query), *i.e.*, ReferFormer (Wu et al., 2022) and MTTR (Botach et al., 2022), and leverage Wav2Vec (same as our speech encoder) (Baevski

| Method | Query | $\mathcal{J}\&\mathcal{F}$ | $\mathcal{J}$ | $\mathcal{F}$ |
|---|---|---|---|---|
| Text-referring Video Object Segmentation (for reference) | | | | |
| ReferFormer | Text | 67.2 | 66.3 | 68.1 |
| MTTR | Text | 64.0 | 63.2 | 64.7 |
| Speech-referring Video Object Segmentation | | | | |
| ReferFormer | Text (ASR) | 58.4 | 56.1 | 60.7 |
| MTTR | Text (ASR) | 54.6 | 52.4 | 56.8 |
| **STBridge (Ours)** | Speech | **65.5** | **63.7** | **67.4** |

Table 2: **Quantitative results for speech-referring object segmentation in videos.** Both the $\mathcal{J}\uparrow$ and $\mathcal{F}\uparrow$ scores are the larger the better. $\mathcal{J}\&\mathcal{F}$ is the average of $\mathcal{J}$ and $\mathcal{F}$ as convention.

et al., 2020) to conduct ASR to adapt them to speech input. We notice that ASR-converted text will degrade the baseline models' performance even without noise impact. We consider this can result from word errors in the converted text from speech. For example, if the target object 'cat' is wrongly recognized as 'cap' by ASR, the R-VOS model will inevitably segment the wrong object.

**Segmentation with noisy speech.** As shown in Table 1, we compare the performance of STBridge to previous text-queried methods with noisy speech as inputs. We modify the signal-noise ratio (SNR) of noisy speech to comprehensively evaluate the noise influence. We notice that ASR-based methods suffer severe performance drops compared to clean speech. In contrast, STBrigde shows a more robust performance with only slight degradation when noise becomes loud.

### 4.4 Visualization

In Fig. 4, we show the qualitative comparisons between our method, i.e., STBridge, and a cascade of Wav2Vec2 (Baevski et al., 2020) (ASR model) and ReferFormer (Wu et al., 2022) (RVOS model). Note that the ASR model is fairly chosen to have

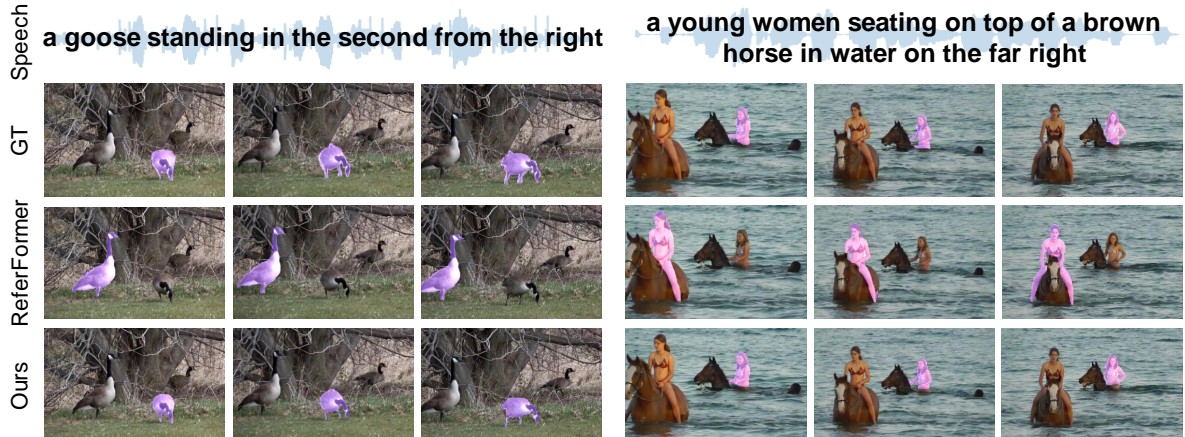

Figure 4: Qualitative comparison between STBridge (our speech-referring VOS model) and ASR-assisted Refer-Former (an assembly of ASR (Baevski et al., 2020) and text-referring VOS (Wu et al., 2022) models).

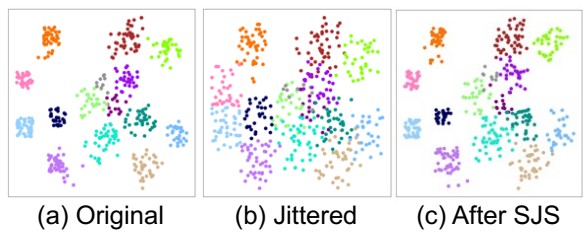

| (a) Original | (b) Jittered | (c) After SJS |

Figure 5: (a) Original embedding. (b) Semantic-jitter-injected embedding. (c) Jitter-suppressed embedding.

the same speech encoder as STBridge. Notably, our method successfully refers to the correct object while ASR-assisted ReferFormer fails to understand the speech input and predicts wrongly.

Fig. 5 demonstrates the function of the SJS module. By comparing Fig. 5 (b) and Fig. 5 (c), we can notice that SJS module effectively suppresses noises in the jittered embeddings.

## 4.5 Ablation Experiments

We conduct ablation studies to show the impact of different modules. Unless otherwise specified, all experiments are conducted on the AVOS test set.

**Module effectiveness.** We conduct experiments to validate the effectiveness of our proposed modules. We add the proposed modules step-by-step as shown in Table 3. (1) With semantic alignment between textual and speech representation, STBridge achieves 61.0 and 55.7 $\mathcal{J}\&\mathcal{F}$ with clean and noisy speech queries, respectively. (2) After equipping the SJS module, STBridge can better handle noises thus boosting the performance with noisy speech to 59.7 $\mathcal{J}\&\mathcal{F}$ while only marginal improvement is achieved with clean speech queries. (3) The

equipping of NSA module benefits the performance with both clean and noisy speech queries. We consider the reason is that the two types of attention in NSA better filter out irrelevant features and help the speech embedding focus on the target object. With all modules, STBridge achieves 65.6 and 62.4 $\mathcal{J}\&\mathcal{F}$ for clean and noisy speech correspondingly.

**Semantic jitter type.** During training, STBridge generates semantic jitter and then learns to suppress it using the SJS module to enhance the noise-tolerant capability of the R-VOS model. We conduct an ablation study to investigate the influence of different semantic jitter types. Specifically, the implemented semantic jitter on the text embedding $g_t$ has a form of $m \circ g_t + \delta$, where $m$ and $\delta$ are binary mask and random noise respectively. As shown in Table 4, we notice masking among both the word-level and channel-level shows an improvement in performance. Random noise $\delta$ brings an additional 0.5 $\mathcal{J}\&\mathcal{F}$ to the final performance.

**Design choices.** We conduct experiments to ablate the design choices in STBridge and their impacts on the segmentation performance. (1) We first study the effect of frame window size selected from the entire video sequence during training. We notice that the window size only shows a marginal impact on the performance, which can be due to the visual encoder and mask decoder being frozen during training. As shown in Table 5a, we notice a window size of 5 achieves the best performance. (2) After that, we ablate on the loss type for semantic alignment in Table 5b. We leverage $L_1$, $L_2$, and Cosine loss to align the text and speech embeddings. We notice that $L_2$ loss achieves the best perfor-

| Module | | | Clean | Noisy |
|---|---|---|---|---|
| SA | SJS | NSA | $\mathcal{J}\&\mathcal{F}$ | $\mathcal{J}\&\mathcal{F}$ |
| ✓ | | | 61.0 | 55.7 |
| ✓ | ✓ | | 61.3 | 59.7 |
| ✓ | | ✓ | 63.9 | 61.4 |
| ✓ | ✓ | ✓ | **65.5** | **62.4** |

Table 3: **Module effectiveness with clean and noisy speech as queries**. SA: semantic alignment. SJS: semantic jitter suppression. NSA: noise-aware semantic adjustment.

| Semantic Jitter Type | | | $\mathcal{J}\&\mathcal{F}$ |
|---|---|---|---|
| Word Mask | Channel Mask | Random Noise | |
| ✓ | | | 61.2 |
| | ✓ | | 61.4 |
| ✓ | ✓ | | 61.9 |
| ✓ | ✓ | ✓ | **62.4** |

Table 4: **Ablation on semantic jitter types.** We conduct ablations on the impact of created semantic jitter types in the SJS module. We conduct this experiment with noisy speech (10 dB).

| $L_v$ | 1 | 3 | 5 | 7 |
|---|---|---|---|---|
| $\mathcal{J}\&\mathcal{F}$ | 59.4 | 61.7 | **62.4** | 62.3 |

(a) **Window size**.

| Type | $L_1$ | $L_2$ | Cosine |
|---|---|---|---|
| $\mathcal{J}\&\mathcal{F}$ | 62.1 | **62.4** | 60.9 |

(b) **Alignment loss**.

| Ratio | 0.05 | 0.1 | 0.2 |
|---|---|---|---|
| $\mathcal{J}\&\mathcal{F}$ | 61.7 | **62.4** | 60.3 |

(c) **Masking ratio**.

| Amp. | 0.1 | 0.5 | 1 |
|---|---|---|---|
| $\mathcal{J}\&\mathcal{F}$ | 61.1 | **62.4** | 62.0 |

(d) **Noise amplitude**.

Table 5: **Design choices for STBridge.** We report the performance with the noisy speech queries on AVOS test set. (a) We ablate the window size (input frame number) during training. (b) We ablate the semantic alignment loss types. (c) We ablate the making ratio for $m$ in creating semantic jitters. The ratio is calculated by $1 - \frac{\text{sum}(m)}{C \times L_t}$. (d) We ablate the amplitude of the random noise $\delta$. The noise $\delta \sim \text{Uniform}(-\text{Amp}, \text{Amp})$.

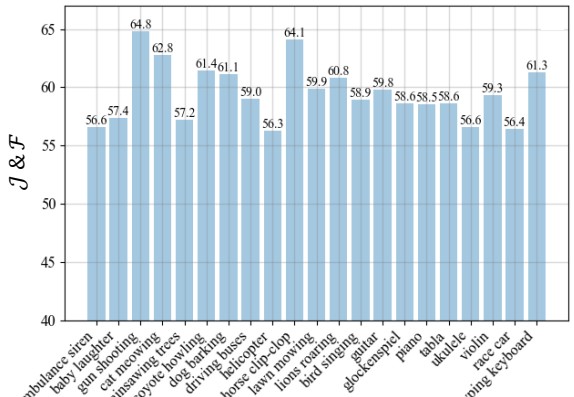

Figure 6: Analysis of the impact of noise categories.

by mixing clean speech from AVOS test set with different categories of audio recordings from Audioset (Gemmeke et al., 2017). As shown in Fig. 6, we illustrate the results of queries with different noise categories. We notice that sustained and loud noises, e.g., ambulance siren, can lead to a severe performance drop compared to short-lived and faint noises, e.g., horse clip-clop.

# 5 Conclusion

In conclusion, this paper presents STBridge, a novel approach that enables R-VOS models trained on clean text-video pairs to adapt to noisy speech as referring guidance, maintaining robust performance. The approach incorporates semantic jitter suppression (SJS) and noise-aware semantic adjustment (NSA) modules to enhance noise tolerance in speech queries. Experimental results demonstrate the effectiveness of STBridge, outperforming previous methods on three benchmarks. STBridge expands the applicability of R-VOS models, enabling robust speech-referred video object segmentation in real-world scenarios.

**Limitation.** In spite of STBrdge's high performance on existing benchmarks, we only consider the scenario that text and speech queries are in the same language. Bridging text and speech in different languages can impose more challenges as the semantic spaces may suffer more divergence, which will be our future focus.

mance among them. (3) Semantic jitter suppression is an essential component in STBridge for noise tolerance. We conduct ablation studies to demonstrate the impact of different masking ratios and random noise amplitude. Table 5c demonstrates the performance with different masking ratios of $m \in [0, 1]^{C \times L_t}$ (calculated as $1 - \frac{\text{sum}(m)}{C \times L_t}$). Small masking ratios cannot provide enough perturbation to the inputs while large ratios may lose the semantics to the target object. We find a masking ratio of 0.1 is a good trade-off as shown in Table 5c. (4) We ablate the amplitude of noise $\delta$ added as a semantic jitter in Table 5d. We notice that an amplitude of 0.5 leads to the best performance.

**Noise category.** We conduct an experiment to investigate the impact of different noise categories. We additionally synthesize noisy speech queries

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

## A  More Implementation Details

**Training details.**  Following the previous referring segmentation methods (Wu et al., 2022; Botach et al., 2022; Kamath et al., 2021), we leverage loss weight coefficients $\lambda_{dice}$ and $\lambda focal$ to balance Dice (Li et al., 2019) and focal (Lin et al., 2017b) losses in $\mathcal{L}_{mask}$. And $\lambda_{giou}$ and $\lambda_{L1}$ to balance GIoU (Rezatofighi et al., 2019) and L1 losses in $\mathcal{L}_{box}$. During training, we set $\lambda_{noise} = \lambda_{align} = 1$, $\lambda_{conf} = \lambda_{giou} = \lambda_{dice} = 2$, and $\lambda_{L1} = \lambda_{focal} = 5$. We set the layer number in the transformer encoder in $\varphi$ as 3. The frozen ReferFormer (Wu et al., 2022) is pre-trained on Ref-COCO/+/g (Kazemzadeh et al., 2014) for 12 epochs and then finetuned on the AVOS training set while with text queries as input for 6 epochs. For the text, speech, and video triplets used in STBridge training, we only ensure the text and speech describe the same object while the words in the text and speech may differ slightly.

During STBridge training, the noisy speech input is generated by mixing the original clean speech with randomly picked audio, which is injected as noise, from AudioSet with an SNR ranging from 0 to 40. The noise categories include 'ambulance siren',' baby laughter', 'gun shooting', 'cat meowing', 'chainsawing trees', 'coyote howling', 'dog barking', 'driving buses', 'helicopter', 'horse clip-clop', 'lawn mowing', 'lions roaring', 'bird singing', 'guitar', 'glockenspiel', 'piano', 'tabla', 'ukulele', 'violin', 'race car', 'typing keyboard'. The sampling probabilities for each category are the same.

## B  Detailed Structure of Mask Decoder

We demonstrate the detailed mask decoding process in Figure A. Except for the visual feature $f_t$ and object query $q$ as defined in the main paper, we additionally enroll the prototype masks $\{P_t\}_{t=1}^N$ and instance embedding $\{e_t\}_{t=1}^T$. Given the object query $q \in \mathbb{R}^{C \times 1}$ from STBridge, we first repeat it $N$ times to form the input to the transformer decoder TrD where $N$ is the object candidate number (the final output is selected from object candidates based on confidence score). After that, we generate instance embedding $\{e_t\}_{t=1}^T$ for each time step separately using a shared transformer decoder TrD with encoded memory $\{f_t\}_{t=1}^T$ from visual encoder. The instance embedding here encodes the instance information and is leveraged to guide the mask decoding process. The mask prediction

$M_t$ for each time step $t$ is derived by a dynamic convolution between prototype mask $P_t$ and dynamic weights which are learned from instance embedding $e_t$ by two fully connected layers. The prototype masks $\{P_t\}_{t=1}^T$ is generated by feature pyramid network (FPN) (Lin et al., 2017a) with visual feature $\{f_t\}_{t=1}^T$.

## C  Inference Details of R-VOS models

To obtain the final segmentation result, we select the mask (among $N$ candidates) with highest confidence throughout time as:

$$\hat{M}_t = M_{\hat{s},t},$$
$$\hat{s} = \arg\max_i \{S_{i,1} + \cdots + S_{i,T}\}_{i=1}^N \quad (9)$$

where $\{\hat{M}_t\}_{t=1}^T$ is the masks of referred object. $S_{i,t}$ and $M_{i,t}$ represent the $i$-th slot in $S_t$ and $M_t$ respectively. $\hat{s}$ is the slot with the highest confidence to be the target object. Box predictions can help training procedures but are not used during inference.

| Frozen | $\mathcal{J}\&\mathcal{F}$ | $\mathcal{J}$ | $\mathcal{F}$ |
|--------|------|------|------|
| ✗ | 65.5 | 63.7 | 67.4 |
| ✓ | 58.9 | 57.3 | 60.4 |

Table A: Ablation study on updating R-VOS parameters during training.

## D  Training with Trainable R-VOS Model

We conduct additional ablation studies to show the results of training with updating parameters in R-VOS model. As shown in Table A, we notice that updating R-VOS parameters during the adaptation to noisy speech inputs will result in severe performance degradation. We consider this because 1) the information in noisy speech input is not enough to accurately refer to the object resulting in noises in the training process, and 2) the S-VOS dataset is smaller than the R-VOS dataset leading to overfitting.

## E  More Visualization.

As shown in Fig. B, we demonstrate more visualizations of the proposed method. We notice that our method can correctly refer to the target object and help the R-VOS model segment temporally consistent object masks across frames.

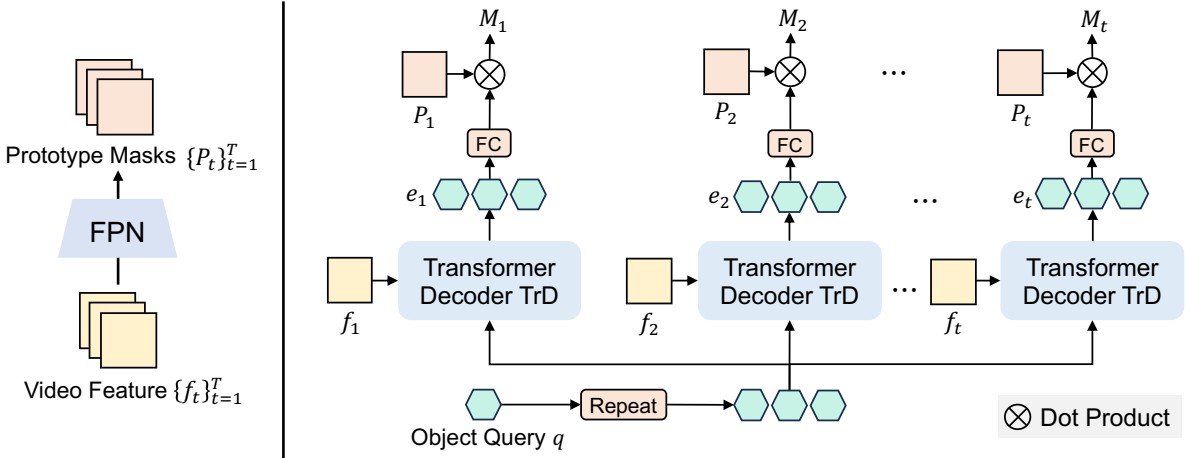

Figure A: Illustration of mask decoder, which derives mask predictions $\{M_t\}_{t=1}^T$ from the visual features $\{f_t\}_{t=1}^T$ the object query $q$. Given the query feature $q$ from STBridge, we first repeat it $N$ times to form the input to the transformer decoder $\mathrm{TrD}$ where $N$ is the object candidate number (the final output is selected from object candidates based on confidence score). After that, we generate instance embedding $\{e_t\}_{t=1}^T$ for each time step separately using a shared transformer decoder $\mathrm{TrD}$ with visual feature $\{f_t\}_{t=1}^T$ from visual encoder. The mask prediction $M_t$ for each time step $t$ is derived by a dynamic convolution between prototype masks $P_t$ and dynamic weights which are learned from instance embedding $e_t$ by two fully connected layers. The prototype masks $\{P_t\}_{t=1}^T$ is generated by feature pyramid network (FPN) (Lin et al., 2017a) with visual feature $\{f_t\}_{t=1}^T$.

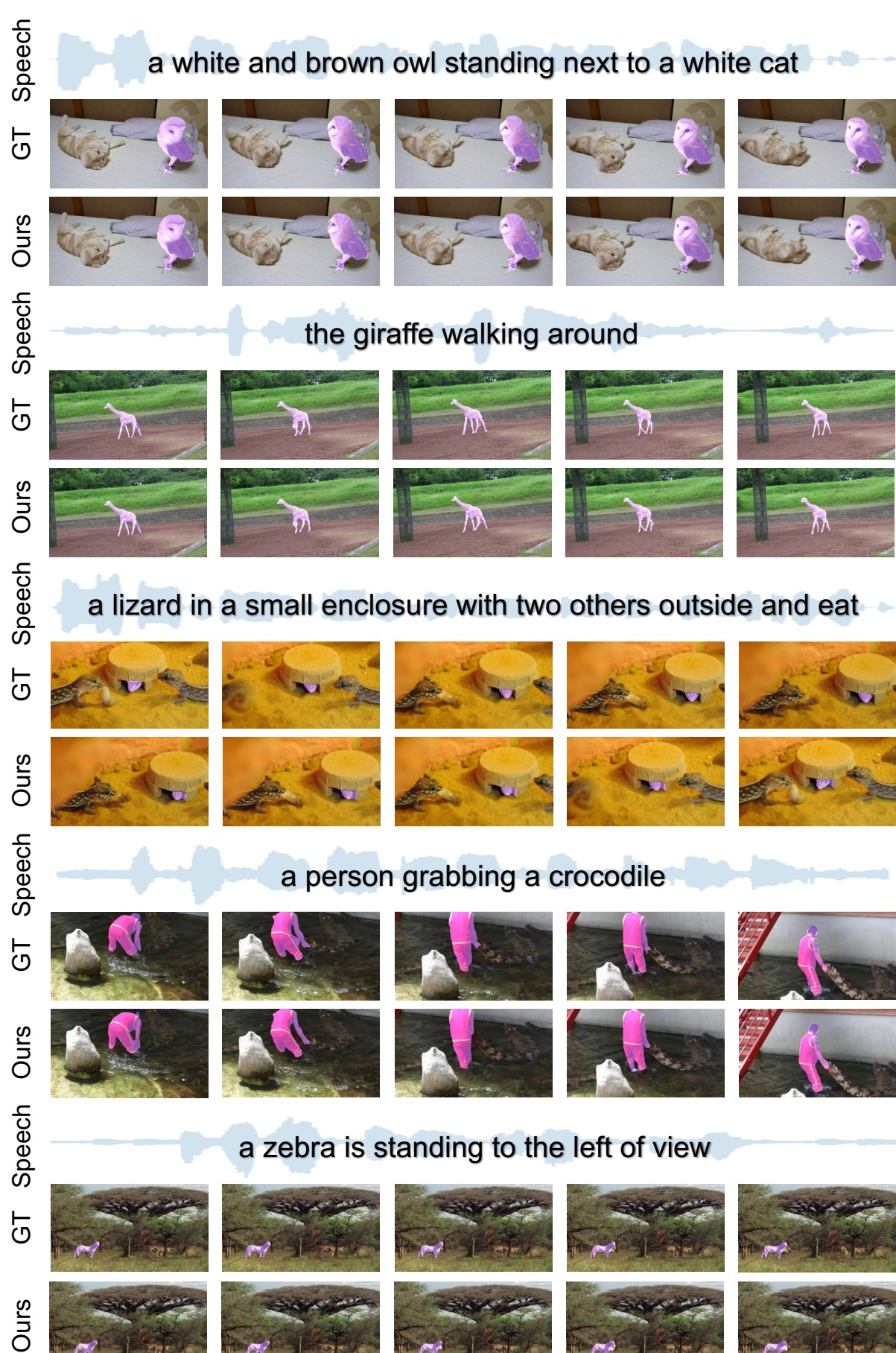

Figure B: More visualization of our method on AVOS test set.