# OpenReview forum: "Towards Noise-Tolerant Speech-Referring Video Object Segmentation: Bridging Speech and Text"
_EMNLP/2023/Conference — EMNLP 2023 Main_

### Official Review · Reviewer_zoVN · 2023-07-21

**Soundness:** 4

**Excitement:**

3: Ambivalent: It has merits (e.g., it reports state-of-the-art results, the idea is nice), but there are key weaknesses (e.g., it describes incremental work), and it can significantly benefit from another round of revision. However, I won't object to accepting it if my co-reviewers champion it.

**Paper Topic And Main Contributions:**

This paper tackles the task of speech query based referring video object segmentation (R-VOS). To handle the noisy nature of speech queries, the authors proposed two modules, namely noise-aware semantic adjustment, and semantic jitter suppression. Experiments show that the proposed modules are effective, and their approach achieves the state-of-the-art performance on three referring video object segmentation benchmarks, outperforming both speech R-VOS models, and ASR + text R-VOS models, closing the gap between speech based models and text based models using ground truth text.


**Questions For The Authors:**

Another benefit of this end2end approach compared to using ASR, is inference speed. I'm curious of the quantitative comparison.

**Reasons To Accept:**

two modules - noise-aware semantic adjustment, and semantic jitter suppression are proposed to handle the noisy speech input for semantic information extraction. Extensive ablation studies showed that the proposed modules are effective. The proposed approach is effective even when no noise is injected (table1). The impact of different noise types are also investigated, and they show that "sustained and loud noises can lead to a severe performance drop compared to short-lived and faint noises"

**Reasons To Reject:**

1. The type of noise is limited to only background noise (and from AudioSet specifically), but the definition of noise in speech is much broader - reverberation, far-field, low bandwidth, missing segments, accents. Accents in particular appeared in the abstract as a motivating factor, but never mentioned in the main text of the paper.
The approach contains a background noise class prediction module, which could mean that the approach only works for background noise. If this is the case, I suggest the authors modify the paper to be clearer on what kind of noise is being tackled here.

2. The generalization ability. Only a limited pool of noise categories (21 kinds in total) are used as injected noise, and therefore not sure how this approach generalize when OOD background noise appear in the speech query.

3. The info regarding the ASR model are largely missing. The only info the author presented in the paper is that it's W2V2, but the size, pretraining/finetuning data are missing. It might be worth considering whether the proposed approach is still competitive compared to a stronger ASR model, e.g. Whisper

**Reproducibility:**

4: Could mostly reproduce the results, but there may be some variation because of sample variance or minor variations in their interpretation of the protocol or method.

**Reviewer Confidence:**

3: Pretty sure, but there's a chance I missed something. Although I have a good feel for this area in general, I did not carefully check the paper's details, e.g., the math, experimental design, or novelty.

---

> ### Author Rebuttal · Authors · 2023-08-28
>
> We thank the reviewer for the time and effort to review our paper. Our answers to the questions are as follows.
>
> ---
>
> **1. The type of noise is limited to only background noise but the definition of noise in speech is much broader. I suggest the authors modify the paper to be clearer on what kind of noise is being tackled here.**
>
> We appreciate your valuable suggestion. Indeed, STBridge is specifically designed to mitigate the impact of background noise in speech. We will make necessary revisions in the paper to explicitly clarify this focus on background noise.
>
> ---
>
> **2. Only a limited pool of noise categories (21 kinds in total) are used as injected noise, and therefore not sure how this approach generalizes when OOD background noise appears in the speech query.**
>
> To evaluate the performance of OOD background noise, we train a model with 21 seen noise categories and test it on 10 randomly picked unseen noise categories from AudioSet. As shown in Tab A, we notice the performance with unseen noise is slightly lower than that with seen noise while still showing obvious superiority compared to training with clear speech (no noise). This indicates the noise-tolerant capability of STBridge can be generalized to unseen noise categories.
>
> |Method|Train Noise| Inference Noise |$\mathcal{J}$\&$\mathcal{F}$|
> |:----|:----:|:----:|:----:|
> |STBridge|NA| Unseen| 55.7 |
> |STBridge|Seen | Seen| 62.4 |
> |STBridge|Seen| Unseen| 61.0 |
>
>
>
> Tab A. STBridge with unseen noise type during inference.
>
> ---
>
> **3. The info regarding the ASR model (size, pretraining/finetuning data) is missing. It might be worth considering whether the proposed approach is still competitive compared to a stronger ASR model, e.g. Whisper.**
>
> We leverage the Wav2Vec2-Base model pre-trained and finetuned on Librispeech 960h as the speech encoder for ASR baselines which shares the same architecture used in STBridge. To investigate the performance with a stronger ASR model, we conduct additional experiments to utilize Whisper-Large as the ASR models as shown in Tab B. We notice that the model with Whisper-Large shows a slightly superior performance than STBridge. However, it's important to highlight that Whisper-Large demands significant computational resources, which in turn negatively affects inference speed, as detailed in Tab C.
>
> |Method|$\mathcal{J}$\&$\mathcal{F}$|
> |:----|:----:|
> |ASR baseline (Wav2Vec2-Base)| 58.4 |
> |ASR baseline (Whisper-Large)| 66.6 |
> |STBridge (Wav2Vec2-Base)| 65.5 |
>
> Tab B. Comparison of models with different ASR models.
>
> ---
>
> **4. Comparison of inference speed between end2end STBridge and baseline using ASR.**
>
> We present the inference speed results for both our end-to-end STBridge and the ASR baseline models in Tab C. Notably, STBridge demonstrates a superior inference speed compared to the ASR baselines. Specifically, when employing the Wav2Vec2-Base model, STBridge outperforms the ASR baseline by 5.0 FPS. Remarkably, in a comparison against the computationally demanding Whisper-Large, STBridge exhibits a substantial improvement of 26.1 FPS.
>
> |Method|FPS|
> |:----|:----:|
> |ASR Baseline (Whisper-Large) | 5.2 |
> |ASR Baseline (Wav2Vec2-Base) | 26.3 |
> |STBridge (Wav2Vec2-Base) | 31.3 |
>
> Tab C. Comparision of model speeds with batchsize=1 on a single NVIDIA P40 GPU. FPS: frames per second.

---

### Official Review · Reviewer_A74t · 2023-07-31

**Soundness:** 4

**Excitement:**

4: Strong: This paper deepens the understanding of some phenomenon or lowers the barriers to an existing research direction.

**Paper Topic And Main Contributions:**

The paper addresses the problem of referring video object segmentation (R-VOS) using speech input, which is more challenging compared to using text input due to noise and potential information loss. The paper makes several contributions to the field of R-VOS and speech-referring segmentation:

Firstly, it addresses the gap between text and speech input, which is an underexplored area in the field. It introduces a method to align the semantic spaces between speech and text, enabling the adaptation of text-input R-VOS models to handle noisy speech input effectively. This allows for a more natural and convenient way for humans to refer to objects in videos using speech.

Secondly, the paper proposes the Noise-Aware Semantic Adjustment (NSA) module, which enhances the R-VOS models to handle incomplete and distorted information from noisy speech. It focuses on extracting accurate semantics from noisy speech, enabling effective segmentation even with perturbed referring queries.

Thirdly, the paper introduces the Semantic Jitter Suppression (SJS) module, which helps the R-VOS models tolerate noisy queries by generating perturbations and suppressing noise in the textual features. This module allows the models to learn from incomplete referring guidance and adapt to noisy speech input.

The paper provides comprehensive experiments on three challenging benchmarks, demonstrating the superiority of the proposed method compared to state-of-the-art approaches. The experiments cover both clean and noisy speech input scenarios, showcasing the noise-tolerant capabilities of the proposed method.

**Questions For The Authors:**

A. How does the proposed method handle cases where the spoken descriptions are ambiguous or inconsistent with the video content? Are there any strategies or techniques to handle such cases?

B. Have you considered using noise reduction techniques to first reduce the noise in speech and then extract text through ASR to perform the R-VOS task? If so, how effective is this approach?

C. How sensitive is the proposed method to variations in speech quality or the levels of noise in the speech? Did you observe any significant performance degradation with extremely noisy speech (e.g. SNR higher than 40 dB) or low-quality recordings?

D. Have you tested the proposed method on languages other than English? If so, what were the challenges and how did the method perform in those cases?

E. Have you considered incorporating other modalities (e.g. visual cues or lip movements) along with speech to improve the performance of the proposed method? If so, what were the challenges and potential benefits of such an approach?

**Reasons To Accept:**

Strengths:
1. Novel Approach: The paper proposes a new approach, STBridge, to bridge the gap between speech and text in R-VOS tasks. The incorporation of two key modules, NSA and SJS, effectively addresses the challenges of noisy speech input and aligning the semantic spaces between speech and text.
2. Robust Performance: The comprehensive experiments conducted on three challenging benchmarks demonstrate that the proposed method outperforms state-of-the-art approaches. STBridge achieves significant improvements in both clean and noisy speech input scenarios, highlighting the effectiveness of the proposed modules.

Benefits to the NLP Community:
1. Expanded Scope: The paper addresses a gap in the existing literature by exploring the underexplored area of speech-input R-VOS models. By adapting text-input R-VOS models to accommodate noisy speech input effectively, the proposed method expands the scope of R-VOS research and opens up new possibilities for multimodal HCI tasks.
2. Noise-Tolerant Speech Understanding: The robustness of the proposed method to handle noisy speech queries provides valuable insights for researchers working on speech understanding tasks. This can contribute to the development of more accurate and reliable speech-based models, benefiting various NLP applications beyond video object segmentation.
3. Practical Applications: The practical applications of the proposed method, such as video editing and augmented reality, have significant implications for the NLP community. The successful adaptation of text-input R-VOS models to handle noisy speech input enables the development of more user-friendly and natural human-computer interaction systems, enhancing the user experience in real-world scenarios.

**Reasons To Reject:**

Weakness:
Lack of analysis of failure cases: The paper focuses on the performance improvements achieved by the proposed method but does not analyze the failure cases where the method might not perform well. Understanding the limitations and failure cases of the approach would provide a more comprehensive evaluation.

Risk:
Lack of real-world scenarios: The evaluation is conducted on benchmark datasets, and it would be beneficial to include real-world scenarios or scenarios with more complex visual scenes. This would provide a more realistic evaluation of the approach.

**Reproducibility:**

3: Could reproduce the results with some difficulty. The settings of parameters are underspecified or subjectively determined; the training/evaluation data are not widely available.

**Reviewer Confidence:**

4: Quite sure. I tried to check the important points carefully. It's unlikely, though conceivable, that I missed something that should affect my ratings.

---

> ### Author Rebuttal · Authors · 2023-08-27
>
> We thank the reviewer for the time and effort to review our paper. Our answers to the questions are as follows.
>
> ---
>
> **1. Lack of analysis of failure cases.**
>
> Thanks for the valuable suggestion. We observe that STBridge faces challenges in robustly segmenting target objects within crowded scenes, particularly when dealing with multiple objects belonging to the same class. We attribute this suboptimal performance to the fact that STBridge relies on a sentence-level embedding (same as the underlying R-VOS model), to capture both entity and contextual information (such as color, position, and action of the object). Consequently, in the case of complex sentences, certain aspects of contextual information might be overlooked during the process of feature extraction.
>
> To improve, we consider it could be beneficial to encode the entity and each contextual information (e.g., color, position, and action) into separate embeddings to better handle complex descriptions. Nevertheless, applying this approach requires retraining the currently frozen R-VOS model to adjust the object query generation, which may impair the current plug-and-play usability of STBridge.
>
> We will consider the potential improvement in our future work. We will include the failure case analysis in our revision.
>
> **2. The evaluation is conducted on benchmark datasets, and it would be beneficial to include real-world scenarios or scenarios with more complex visual scenes.**
>
> We appreciate the reviewer for raising the concern regarding real-world scenarios. We concur that a more extensive evaluation with more complex scenes would be valuable. Nevertheless, due to the substantial manual effort required for collecting ground-truth masks for evaluating video segmentation, we were regrettably unable to carry out this evaluation during the rebuttal period. We will keep the constructive suggestion in mind and include the suggested evaluation in our future work.
>
> ---
>
> **3. How does the proposed method handle cases where the spoken descriptions are ambiguous or inconsistent with the video content? Are there any strategies or techniques to handle such cases?**
>
> As STBridge aims to extend a frozen R-VOS model with noisy speech-referring capability, its behavior with ambiguous or inconsistent referring guidance can depend on the underlying R-VOS model.
>
> We notice that a recent study $\mathrm{R}^2$-VOS proposes a cyclic structure consistency to detect and filter out the false positives caused by inconsistent text-video pairs in R-VOS task (no output when the referred object does not exist in the video). By applying STBridge to the $\mathrm{R}^2$-VOS, we believe that ambiguous and inconsistent spoken descriptions can be successfully tackled.
>
> [R1] Robust Referring Video Object Segmentation with Cyclic Structural Consensus, ICCV 2023
>
> ---
>
> **4. Have you considered using noise reduction techniques to first reduce the noise in speech and then extract text through ASR to perform the R-VOS task? If so, how effective is this approach?**
>
> Thanks for your suggestion. We conduct an additional comparison between STBridge and ASR baseline equipped with a popular speech enhancement method, SepFormer [R2]. As shown in Tab A, we notice that even when assisted with speech enhancement, the ASR baseline is still suppressed by STBridge. As shown in Tab 2, when no noise is added, STBridge still outperforms ASR baselines which indicates the upper bound of utilizing speech enhancement.
>
> |Method|$\mathcal{J}$\&$\mathcal{F}$|
> |:----:|:----:|
> |ASR| 53.4|
> |ASR+SE| 58.0|
> |STBridge|62.4|
>
> Table A. Comparision with ASR and ASR+speech enhancement (SE) baselines under 10dB noise.
>
> [R2] https://huggingface.co/speechbrain/sepformer-wham16k-enhancement
>
> ---
>
> **5. How sensitive is the proposed method to variations in speech quality or the levels of noise in the speech? Did you observe any significant performance degradation with extremely noisy speech or low-quality recordings?**
>
> We ablate on the different SNR of noise as shown in Tab 2. The lower the SNR the larger the noise. Compared to no noise, the performance with 10 dB noise drops 3.2 $\mathcal{J}$\&$\mathcal{F}$. We conduct an additional experiment with more noisy speech (5 dB), and the results drop significantly by 23.5 $\mathcal{J}$\&$\mathcal{F}$ compared to no noise which indicates the speech encoder fails to understand the content in such a noise level and the model randomly segment objects without a clear guidance.
>
> ---
>
> **6. Have you tested the proposed method on languages other than English? If so, what were the challenges and how did the method perform in those cases?**
>
> Regrettably, we have not been able to conduct such tests due to the absence of publicly available datasets catering to other languages. It's important to note that STBridge is not tailored exclusively to English, and its design allows for potential applicability in different languages.
>
> Nonetheless, we recognize that if the R-VOS model is primarily trained in English (as is commonly the case) while the target speech is in a different language, it could present challenges in aligning the distinct speech and text spaces. The divergence between the English text space and the speech characteristics of other languages may pose hurdles to effective alignment.
>
> ---
>
> **7. Have you considered incorporating other modalities (e.g. visual cues or lip movements) along with speech to improve the performance of the proposed method? If so, what were the challenges and potential benefits of such an approach?**
>
> The integration of other modalities, such as visual cues or lip movements, in conjunction with speech, represents a promising avenue for enhancing the performance of speech-referring VOS. Here, we outline the potential benefits and challenges associated with this approach:
>
> - Benefits:  Visual cues, encompassing facial expressions, gaze, and lip movements, can provide supplementary information alongside speech, aiding in disambiguation during dialogue, particularly in noisy environments. We believe that the inclusion of visual cues could benefit speech-referring VOS by enhancing speech comprehension, thereby enabling the extraction of more precise object queries for segmentation.
>
> - Challenges: While visual cues offer complementary data, effectively leveraging them to facilitate speech comprehension remains an open challenge. Specific challenges include:
>
>     - Handling Incomplete Visual Observations: Given that facial features, including lips, can be partially obscured or occluded, devising strategies to deal with incomplete or partial visual observations is crucial when extracting visual features.
>
>     - Balancing Speech and Visual Information: Striking the right balance between relying on speech and incorporating visual cues without overemphasizing one over the other is a key challenge to address.
>
>     - Temporal Alignment: Given that temporal information is critical for speech comprehension, aligning visual and audio features presents an important consideration.
>
> We will include the discussion in our revision.
>
> ---
>
> We would like to thank the reviewer for the valuable comments which deeply inspire us for future research.

---

### Official Review · Reviewer_uQBw · 2023-08-07

**Soundness:** 4

**Excitement:**

4: Strong: This paper deepens the understanding of some phenomenon or lowers the barriers to an existing research direction.

**Justification For Ethical Concerns:**

No concerns.

**Paper Topic And Main Contributions:**

This paper proposed a novel approach that adapts R-VOS models trained on clean text-video pairs to noisy speech input.  To align the semantic spaces between speech and text by incorporating two key modules: 1) Noise-Aware Semantic Adjustment for clear semantics
extraction from noisy speech; and 2) Semantic Jitter Suppression enabling R-VOS models to tolerate noisy queries.


**Questions For The Authors:**

1. What are the essential difference between noise in query and noise in ASR (line 067 vs line 076). From my understanding, the first noise is due to ASR errors which is mostly caused by the second noise.
2. Following the above question, can you leverage the background noise from audio signals as contextual information for better queries?
3. Is it possible to extend your idea to ASR-free queries, i.e., taking raw audio signal as input to the R-VOS model?
4. Can you use entity/keywords extraction from speech instead of ASR?
5. The approach relies on training data with the triplet set of video, text and audio, all synced. Is there enough data like this? How could you leverage the data with only aligned video & audio, or text and audio, etc?

**Reasons To Accept:**

1. The method provides solution to the important field of speech driven video object segmentation with many application scenarios.
2. The novelty is sufficient though the solution is a modularized approach which provides step-by-step functions to deal with a challenging task. Hopefully this could lead to a more end-to-end solution from audio signals to VOS.
3. The experiments are thorough with good results and ablation studies.
4. The reference looks comprehensive to me.
5. The writing is clear and easy to follow.

**Reasons To Reject:**

Please refer to my questions below for possible improvement, mostly in the approach, or at least in the writing to clarify a few details.

**Reproducibility:**

4: Could mostly reproduce the results, but there may be some variation because of sample variance or minor variations in their interpretation of the protocol or method.

**Reviewer Confidence:**

4: Quite sure. I tried to check the important points carefully. It's unlikely, though conceivable, that I missed something that should affect my ratings.

---

> ### Author Rebuttal · Authors · 2023-08-26
>
> We thank the reviewer for the time and effort to review our paper. Our answers to the questions are as follows.
>
> ---
>
> **1. What is the essential difference between "noise in query" and "noise in ASR" (Line 067/Line 076)? From my understanding, the first noise is due to ASR errors which are mostly caused by the second noise.**
>
> We agree that the "noise in query" is mostly caused by "noise in ASR". We separately analyze them because they occur in different components of the model and can be addressed differently. We summarize the differences as follows:
>
> |Noise|Module|Our Solution|
> |:----| :----| :----|
> |in language query|Frozen R-VOS model|Adapt R-VOS model to tolerate noisy queries (Line 075 -> SJS module)|
> |in speech understanding|Speech encoder|Enhance speech encoder to extract more accurate embedding (Line 082 -> NSA module)|
>
> We will highlight the differences in the revision.
>
> ---
>
> **2. Can you leverage the background noise from audio signals as contextual information for better queries?**
>
> We consider that the background noise has been incorporated as contextual information in the noise-aware semantic adjustment module (NSA). As shown in Fig 3, we first extract a noise embedding $g_n$ which encodes the background noise information and then learn dynamic kernels from it to modulate the noisy speech embedding $g_{ns}$. In this way, we can incorporate the background noise as contextual information to enhance speech query learning. As shown in Tab 3, NSA module brings 2.9 $\mathcal{J}$\&$\mathcal{F}$ and 5.7 $\mathcal{J}$\&$\mathcal{F}$ gain with clean and noisy speech queries respectively.
>
> ---
>
> **3. Is it possible to extend your idea to ASR-free queries? Can you use entity/keyword extraction from speech instead of ASR?**
>
> We would like to clarify that STBridge is an ASR-free method that enables end-to-end speech-referring VOS. As shown in Fig 2, we apply the semantic alignment constraint between speech embedding $g_s$ and text embedding $g_t$ during training. In this way, we can directly obtain the object query $q$ from the aligned speech embedding during inference.
>
> We recognize that it might be misleading to use an example of an ASR-based solution to introduce the challenges (Line 063). We will revise the wording to enhance clearance in the revision.
>
> ---
>
> **4. The approach relies on a triplet set of video, text, and audio. Is there enough data like this? How could you leverage the data with only aligned video \& audio or text and audio?**
>
> In this work, we utilize three popular datasets containing 18,811 videos with more than 150,000 frames. For each video, a text and a speech describing the target object are included. Given this extensive dataset, we consider it sufficient to demonstrate the effectiveness of the proposed method.
>
> To further scale up for practical consideration, data augmentation is necessary to create the required data triplet. When only video & audio pairs are available, we consider the text can be first generated by off-the-shelf ASR model and then train STBridge with the video-audio-text triplets. For the scenario with only audio & text pairs, we consider the proposed semantic alignment constraint can still be leveraged. Specifically, the speech encoder can be trained by distilling knowledge from the frozen text encoder. We notice several multimodal foundation models in a variety of fields (such as captioning, text-to-speech, and segmentation) have been proposed recently. By leveraging those powerful foundation models to create pseudo labels, training with limited data access becomes feasible.

---

### Meta-Review · Area_Chair_FcyG · 2023-09-15

**Recommendation:** 5

**Metareview:**

This paper presents a novel approach that enables referring video object segmentation (R-VOS) models trained on clean text-video pairs to adapt to noisy speech as referring guidance and maintain robust performance. The authors achieve this by using Noise-Aware Semantic Adjustment for precise semantics extraction from noisy speech without using ASR and Semantic Jitter Suppression, which makes R-VOS models more robust to noisy queries. Experiments on three challenging benchmarks demonstrate the superiority of the proposed method compared to state-of-the-art approaches.

The reviewers praised the novelty of the proposed solution, many practical applications, clear writing and robust experimental analysis. The authors successfully answered all of the reviewers' questions during the rebuttal.

---

### Decision · Program_Chairs · 2023-10-07

**Decision:**

Accept-Main

**Comment:**

This paper presents a novel approach that enables referring video object segmentation (R-VOS) models trained on clean text-video pairs to adapt to noisy speech as referring guidance and maintain robust performance. The authors achieve this by using Noise-Aware Semantic Adjustment for precise semantics extraction from noisy speech without using ASR and Semantic Jitter Suppression, which makes R-VOS models more robust to noisy queries. Experiments on three challenging benchmarks demonstrate the superiority of the proposed method compared to state-of-the-art approaches.

The reviewers praised the novelty of the proposed solution, many practical applications, clear writing and robust experimental analysis. The authors successfully answered all of the reviewers' questions during the rebuttal.